# Infraspinatus Fascial Dysfunction as a Cause of Painful Anterior Shoulder Snapping: Its Visualization via Dynamic Ultrasound and Its Resolution via Diagnostic Ultrasound-Guided Injection

**DOI:** 10.3390/diagnostics13152601

**Published:** 2023-08-04

**Authors:** King Hei Stanley Lam, Daniel Chiung Jui Su, Yung-Tsan Wu, Mario Fajardo Pérez, Kenneth Dean Reeves, Philip Peng, Bradley Fullerton

**Affiliations:** 1The Department of Clinical Research, The Hong Kong Institute of Musculoskeletal Medicine, Hong Kong; 2Faculty of Medicine, The Chinese University of Hong Kong, Hong Kong; 3Faculty of Medicine, The University of Hong Kong, Hong Kong; 4Center for Regional Anesthesia and Pain Medicine, Wan Fang Hospital, Taipei Medical University, Taipei 110, Taiwan; 5Center for Regional Anesthesia and Pain Medicine, Chung Shan Medical University Hospital, Taichung 402, Taiwan; 6Department of Physical Medicine and Rehabilitation, Chi Mei Medical Center, Tainan 710, Taiwan; dr.daniel@gmail.com; 7Department of Physical Medicine and Rehabilitation, Tri-Service General Hospital, School of Medicine, National Defense Medical Center, Taipei 114, Taiwan; crwu98@gmail.com; 8Integrated Pain Management Center, Tri-Service General Hospital, School of Medicine, National Defense Medical Center, Taipei 114, Taiwan; 9Department of Research and Development, School of Medicine, National Defense Medical Center, Taipei 114, Taiwan; 10Ultradissection Group, Calle Arturo Duperier, 28029 Madrid, Spain; mfajardoperez@yahoo.es; 11Morphological Madrid Research Center, Calle Arturo Duperier, 28029 Madrid, Spain; 12Vithas Hospital, Calle Santa Fe 12, Río Odiel 14,8, cp Móstoles, 28935 Madrid, Spain; 13Private Practice PM&R and Pain Management, 4840 El Monte, Roeland Park, KS 66205, USA; deanreevesmd@gmail.com; 14Department of Anesthesiology and Pain Medicine, University of Toronto, Toronto, ON M5T 2S8, Canada; philip.peng@uhn.ca; 15Private Practice: ProloAustin, 2714 Bee Cave Road, Suite 106, Austin, TX 78746, USA; drbdf@aol.com; 16Texas A&M College of Medicine, 3950 North A.W. Grimes Boulevard, Round Rock, TX 78665, USA

**Keywords:** painful shoulder snapping, dynamic ultrasonography, infraspinatus fascia, scapulothoracic dyskinesia

## Abstract

This report presents the first case of painful anterior shoulder snapping due to a thickened, fibrotic bursa snapping between the subscapularis and the short head of the bicep during external and internal rotation of the humerus. A 46-year-old presented with a 10-month history of on-and-off anterolateral right shoulder pain and snapping. Direct treatment to the anterior suspected lesions partially and temporarily relieved the pain but did not reduce the snapping. Further musculoskeletal examination and dynamic ultrasound scanning showed dysfunction in the scapulothoracic movement and defects of the muscles that interact with the infraspinatus aponeurotic fascia. An ultrasound-guided diagnostic injection to the suspected lesions in the infraspinatus fascia and its muscles attachments improved the scapulothoracic movement, and the snapping and pain were eliminated immediately after the injection, which further shows that the defects in the infraspinatus fascia may be the root cause of the painful anterolateral snapping. The importance of the infraspinatus fascia and its related muscle in maintaining the harmony of the scapulothoracic movement and flexibility of the shoulder is considerable.

**Figure 1 diagnostics-13-02601-f001:**
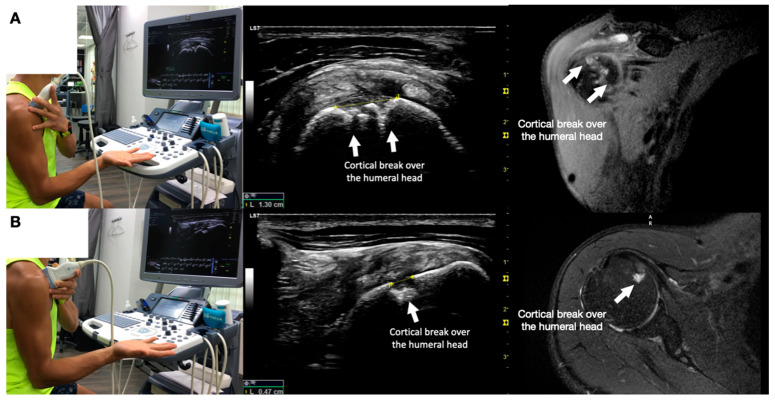
Shows the sonographic appearance of a cortical break over the humeral head deep in the subscapularis tendon and its corresponding magnetic resonance image appearance. The white arrows pointed to the cortical break over the humeral head. (**A**) Shows the view of the break in its long axis. (**B**) Shows the short axis of the cortical break. This report aimed to present a case of painful anterior shoulder snapping, apparently due to the compression of thickened fibrotic bursa between the subscapularis and the short head of the bicep during external and internal rotation of the humerus. However, an ultrasound-guided injection of the thickened and fibrotic bursa did not reproduce the pain or improve snapping. Ultrasound scanning identified a cortical break of the humeral head underneath the subscapularis as a potential mechanical effect of snapping. Nevertheless, an ultrasound-guided injection of the cortical break reproduced and temporarily reduced the pain with snapping and residual anterior shoulder pain, but it did not alter the snapping or weakness in the right shoulder flexion with the humerus in external rotation. A 46-year-old musculoskeletal physician with a 10-month history of on-and-off anterolateral right shoulder pain and snapping [1,2] presented a numerical rating scale (NRS) score of 6–8/10. He had noticed the anterior right shoulder nonpainful snapping for several years and with recent aggravation. Substantial dysfunction and impairment were evidenced by 80% pain and 70% disability sub-scores on the Shoulder Pain and Disability Index (SPADI) [3], with a total score of 73.8% out of a maximum severity of 100. His pain and dysfunction were unresponsive to all manner of conservative musculoskeletal treatments, which substantially affected his sleep and vocation as a musculoskeletal physician. Snapping was prominent during internal and external rotation of the glenohumeral joint at different degrees of abduction and/or flexion. Physical examination revealed a normal range of motion of the shoulder. Tenderness was observed over the anterolateral part of the shoulder over the lesser tuberosity. Internal and external rotation of the shoulder revealed snapping over his anterolateral shoulder, and the most severe snapping occurred when he flexed and abducted his shoulder to 90°, followed by internal and external rotations. Resisted shoulder flexion power tested by the treating physician with the elbow straight and the shoulder flexed at 90° with the internally rotated humerus (thumb down, similar to the empty can sign), both straight and abducted shoulder, had normal power and no pain. However, the power of resisted shoulder flexion with the elbow straight and the shoulder flexed to 90° with the humerus externally rotated (palm up) was diminished and associated with pain (Appendix A). Dynamic ultrasound scanning of the painful anterior shoulder snapping during internal and external rotation was shown in Appendix A. The cortical break underneath the subscapularis, suspected due to the mechanical effects of the snapping and causing the anterior shoulder pain, was demonstrated in Figure 1. Ultrasound-guided digital palpation of the cortical break reproduced the patient’s usual pain in the anterolateral shoulder.

**Figure 2 diagnostics-13-02601-f002:**
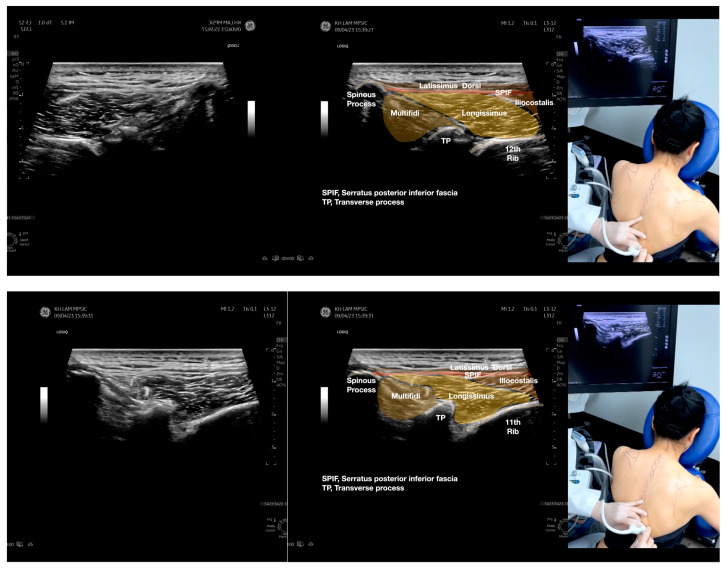
Sonoanatomy of the lateral border of the inferior trapezius, its related muscles, and the infraspinatus fascia. The step-by-step scanning techniques of the structures illustrated in this figure have been shown in Appendix A. Available online: https://www.dropbox.com/s/vjj49wiwdsidouz/Figure%202.docx?dl=0 (accessed on 1 January 2023). Dynamic ultrasound scanning of the anterior shoulder with the humerus externally rotated (palm up), elbow straight, and the shoulder actively flexed and abducted at about 90 to 100 degrees shows that the fibrotic and thickened bursa was noted to be snapping between the subscapularis tendon and the coracoacromial ligament, as shown in Appendix A. The patient noticed this snapping during many of his daily activities with his shoulder flexed just above 90 degrees, e.g., taking off clothes and combing hair. Ultrasound-guided injection of the thickened and fibrotic bursa did not reproduce the usual pain nor reduce the pain, and it did not improve the shoulder snapping and flexion power with the elbow straight, shoulder flexed to 90°, and the humerus externally rotated (palm up). Ultrasound-guided injection of the cortical break reproduced the concordant pain and temporarily and partially reduced the pain with snapping and residual anterior shoulder pain (Appendix A), but it did not change the snapping or weakness in the right shoulder flexion with the humerus in external rotation. With the failure of direct anterior treatment to the suspected lesions, it was essential to further explore other causes for the anterior painful snapping. Further detailed musculoskeletal examination of the scapular movement showed that the right scapulothoracic movements were not smooth compared to the left counterpart (Appendix A) [4,5]. There might be some disruptions in the right infraspinatus fascia (IF) and its related muscles, including the lateral edge of the right inferior trapezius, rhomboid minor and major [6], teres major, latissimus dorsi [7,8] (LD), and posterior deltoid attachments to the IF [9,10]. Clinically, prominent and active trigger points were observed in the right infraspinatus muscle. Holding the inferior angle of the scapula, with the examiner’s fingers, especially over the origin of the teres major muscle over the scapula, the latissimus dorsi [11,12,13] and the scapular insertion of the rhomboid major muscle significantly improved the power of the resisted shoulder flexion with the elbow straight and the shoulder flexed to 90° with the humerus externally rotated (palm up). In order to illustrate the utilization of ultrasound-guided sonopalpation and ultrasound-guided digital palpation for detecting the suspected lesions of the IF and its related muscles, we presented the following videos (Appendix A) and still images (Figure 2, Figure 3, Figure 4 and Figure 5) to demonstrate the essential techniques and crucial points of getting the normal sonoanatomy of the IF and its related structures. The gross anatomy of the IF and its related muscles were also shown in Figure 6, Figure 7 and Figure 8.

**Figure 3 diagnostics-13-02601-f003:**
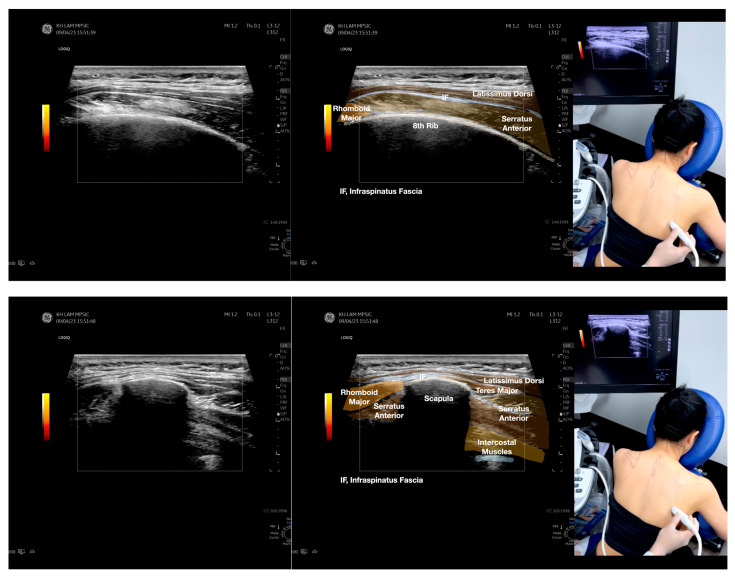
Sonoanatomy of the lateral boarder of the scapular, its related muscles, and the infraspinatus fascia. Appendix A has demonstrated the step-by-step scanning techniques of these structures shown in this figure. Available online: https://www.dropbox.com/s/891m19vvw6bak34/Figure%203.docx?dl=0 (accessed on 1 January 2023).

**Figure 4 diagnostics-13-02601-f004:**
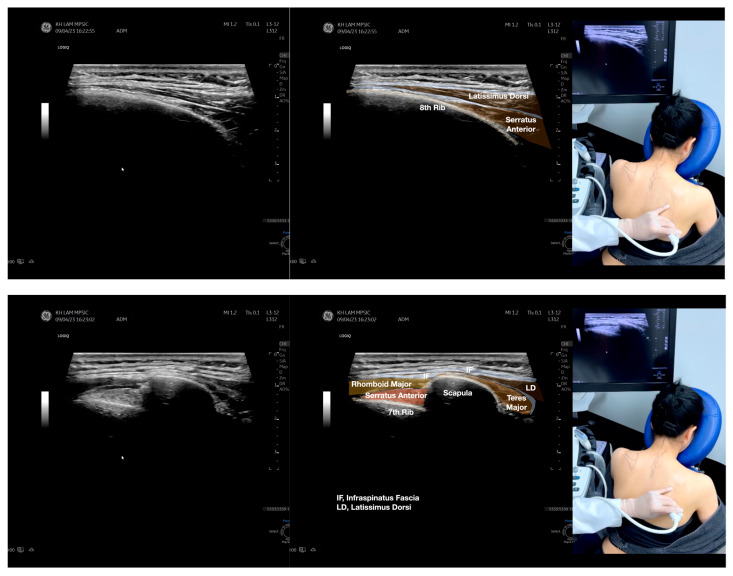
Sonoanatomy of the medial edge of the scapula, its related muscles, and the infraspinatus fascia. The step-by-step scanning techniques of these structures illustrated in this figure have been shown in Appendix A. Available online: https://www.dropbox.com/s/iaej3rxhl83kqt4/Figure%204.docx?dl=0 (accessed on 1 January 2023).

**Figure 5 diagnostics-13-02601-f005:**
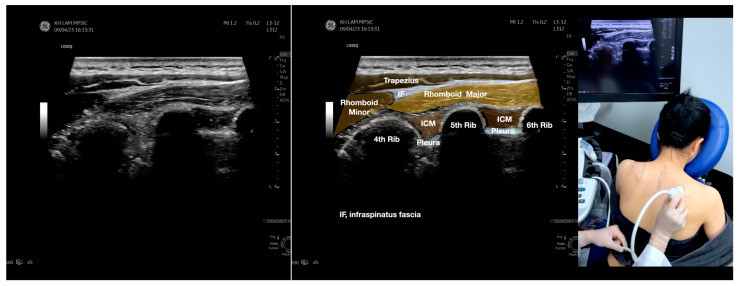
Sonoanatomy of scanning the infraspinatus fascia in the sagittal plane from the rhomboid minor laterally, then back to the rhomboid major. The step-by-step scanning techniques of the structures illustrated in this figure have been shown in Appendix A. Available online: https://www.dropbox.com/s/gu9hhrdq9erin6n/Figure%205.docx?dl=0 (accessed on 1 January 2023).

**Figure 6 diagnostics-13-02601-f006:**
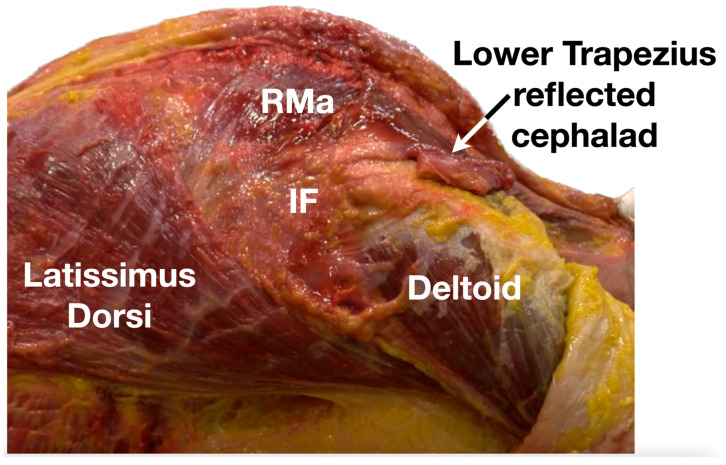
The IF forms the roof of the infraspinatus fossa and is an aponeurotic fascia that extends from the scapular spine to the superior border of the LD and from the medial to lateral borders of the scapula. It contains the infraspinatus and teres minor muscles, but not the teres major. It separates the teres minor from the infraspinatus and teres major by septum-like fibers. It also serves as an additional attachment for the origin of other muscle fibers [9] providing important mechanical kinetics and stability functions. Medially, the IF is located deep to the lower trapezius muscle, which attaches to the medial scapular spine and blends with fascia from the rhomboid major. Laterally, the IF is deep to the posterior belly of the deltoid muscle and is an important attachment for the transmission of forces from the posterior belly of the deltoid to the scapular spine [9]. All these fascial bands work closely with the related muscles, namely the posterior belly of the deltoid, mid and lower trapezius, rhomboid major and minor, infraspinatus, teres minor and major, and LD to maintain a smooth and coherent scapulothoracic movement [4,5,6,7,8,9,10,11,12,13]. Any disruption in any bands of the IF or the muscles attached to the IF can lead to scapulothoracic dyskinesia, and the ipsilateral scapula in a more anterior or protracted state can lead to anterior shoulder impingement and subsequent snapping. IF, infraspinatus fascia; LD, latissimus dorsi; RMa, rhomboid major. Available online: https://www.dropbox.com/s/3c5a3eqd3bc1qpr/Figure%206.docx?dl=0 (accessed on 1 January 2023).

**Figure 7 diagnostics-13-02601-f007:**
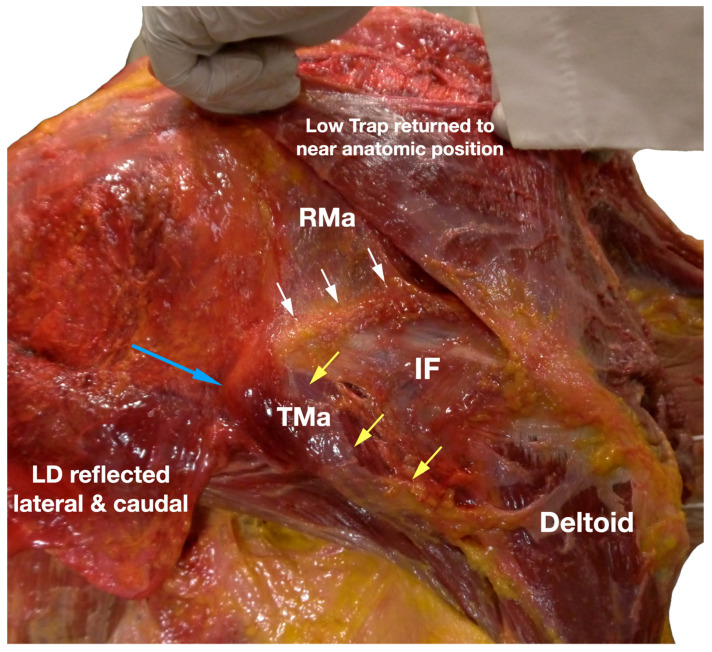
This picture showed an extensive continuation of the different fasciae of the muscles with each other. When the lower trapezius has been returned to a near anatomic position, the white arrows in this picture showed the continuity of the IF to the rhomboid major fascia; the yellow arrows showed the continuation of the IF to the teres major fascia; and the blue arrow points to the inferior angle of the scapula where the rhomboid major and the serratus anterior blend together, and the latissimus dorsi fascia blends with the teres major fascia [10]. IF, infraspinatus fascia; LD, latissimus dorsi; RMa, rhomboid major; TMa, teres major; Trap, trapezius. Available online: https://www.dropbox.com/s/3wxcocclhasfu53/Figure%207.docx?dl=0 (accessed on 1 January 2023).

**Figure 8 diagnostics-13-02601-f008:**
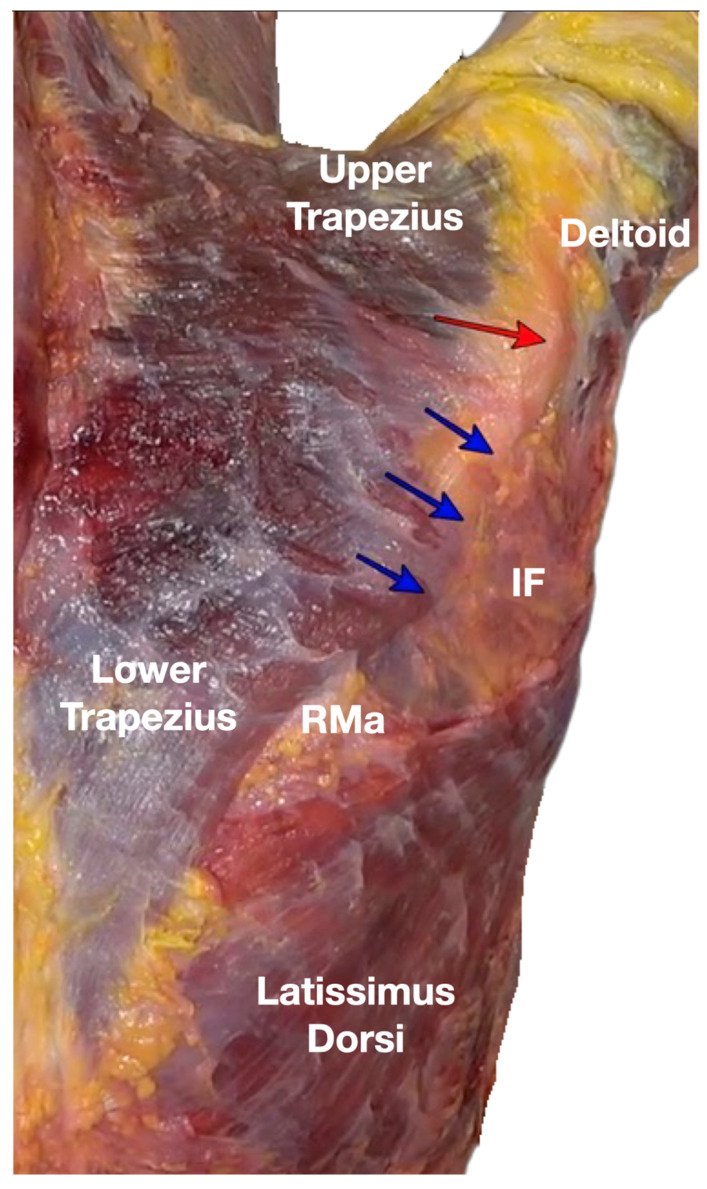
This picture showed the continuation of the trapezius fasciae with the fasciae from latissimus dorsi, rhomboid major and deltoid. The blue arrow showed the continuity of the lower trapezius and its fascia with the IF. The red arrow depicted the insertion of the lower trapezius tendon into the scapular spine. IF, infraspinatus fascia; RMa, rhomboid major. Available online: https://www.dropbox.com/s/7c6d6j2391epb8w/Figure%208.docx?dl=0 (accessed on 1 January 2023). The IF forms the roof of the infraspinatus fossa and is an aponeurotic fascia that extends from the scapular spine to the superior border of the LD and from the medial to lateral borders of the scapula. It contains the infraspinatus and teres minor muscles, but not the teres major. It separates the teres minor from both the infraspinatus and teres major via septum-like fibers. It also serves as an additional attachment for the origin of other muscle fibers [9], providing important mechanical kinetics and stability functions. Medially, the IF is located deep in the lower trapezius muscle, which attaches to the medial scapular spine and blends with the fascia from the rhomboid major. Laterally, the IF is deep to the posterior belly of the deltoid muscle and is an important attachment for the transmission of forces from the posterior belly of the deltoid to the scapular spine [9]. All these fascial bands work closely with the related muscles, namely the posterior belly of the deltoid, mid and lower trapezius, rhomboid major and minor, infraspinatus, teres minor and major, and LD to maintain a smooth and coherent scapulothoracic movement [4,5,6,7,8,9,10,11,12,13]. Any disruption in any bands of the IF or the muscles attached to the IF can lead to scapulothoracic dyskinesia, and the ipsilateral scapula in a more anterior or protracted state can lead to anterior shoulder impingement and subsequent snapping. Sonopalpation is to use the transducer to evaluate myofascial integrity. The aim of sonopalpation is to demonstrate any loss of tension or pre-stress of the fascia, the IF in this patient (Appendix A), which is usually seen as an excessive displacement of the soft tissues when the transducer is compressed onto the soft tissue and the force is transmitted more easily to the bony attachments of the IF. This may cause the scapular edge to excessively protrude toward the subcutaneous or even the dermal layer underneath the transducer. The decreased tension was interpreted as a disruption of fascia integrity or compromised biotensegrity, as described in the thoracolumbar fascia [14]. There is another alternative technique, called ultrasound-guided digital palpation, which serves to diagnose the loss of tension or the excessive displacement of the fascia, which may signify fascial defects, as illustrated in Appendix A. Both techniques have the advantage of real time comparison with the sound side as a control. In this patient, to prove the suspected lesions in the IF found by ultrasound-guided sonopalpation, where it showed a decreased tension and an increased displacement of the soft tissue under sonopalpation compared to the sound side, and is partly shown in Appendix A, we performed an ultrasound-guided injection of 15% hypertonic-glucose mixed with 0.1% lidocaine for the diagnostic and therapeutic injection. We injected the suspected lesions in the IF over its attachments with the teres major and LD on the inferior edge of the IF (Appendix A); in the lateral edge of the inferior trapezius attachments to the medial side of the IF (Appendix A); and in the superolateral attachment of the IF above the teres minor and major (Appendix A). Hypertonic glucose is the primary injectate of these diagnostic and therapeutic injections, which postulates to induce the wound healing cascade and leads to tissue repair [15,16,17]. After the diagnostic and therapeutic hypertonic glucose injection, the volume of the injectate will temporarily increase the tension around the IF, which temporarily improves the scapular dyskinesia and makes shoulder movement more efficient (Appendix A). In this patient, the shoulder snapping (Appendix A) and his anterolateral shoulder pain were eliminated immediately after the injection of the above muscle attachments to the IF. Besides, the power of the resisted shoulder flexion with the elbow straight and the shoulder flexed to 90° with the humerus externally rotated (palm up) was normalized (Appendix A). Pain was eliminated and function normalized immediately after the first ultrasound-guided injection of 15% dextrose mixed with 0.1% lidocaine into the infraspinatus fascial defects. The total SPADI score improved from 73.8 to 40% at a 6-week post-injection follow-up. A second treatment of the residual fascial defect sites with 15% dextrose mixed with 0.1% lidocaine was administered, which again eliminated pain and resolved the residual functional limitations. The patient received no other treatment, but he has been paying particular attention to any postures which would lead to damage in the IF, e.g., prolong protracting the shoulders during treating his patients manually. The patient was pain-free without any snapping symptoms or functional limitations at 3, 6, 12, and 48 months of follow-up. It is important to point out that pain imitation by diagnostic injection does not confirm that the structure being injected is the source of the pathology. In this case, we suspect but cannot confirm that the cortical break was secondary to the scapulothoracic dyskinesia. Abnormal force transmission to the area due to altered mechanics of the humeral head movement could be responsible for the lesion development and the sensitization of nerves in that area, such that an injection would be painful and imitative. By relieving the dyskinesia, we suggest that abnormal force transmission would have ceased. This case report describes a non-reported scenario of painful anterior snapping shoulder syndrome secondary to scapulothoracic dyskinesia due to the disruption of IF integrity. This finding highlights the importance of treating the IF by treating scapulothoracic dyskinesia. As clinicians, in addition to treating the presenting symptoms and signs of the consequence (in this case, the cortical breaks of the anterior humeral head), it is imperative to look for and treat the root causes of the consequence (the IF lesions which caused scapulothoracic dyskinesia) to provide long-term relief and prevent recurrence.

## Data Availability

Data related to this study has been included in the manuscript.

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
