# Peer review of "Infraspinatus Fascial Dysfunction as a Cause of Painful Anterior Shoulder Snapping: Its Visualization via Dynamic Ultrasound and Its Resolution via Diagnostic Ultrasound-Guided Injection"

_diagnostics, 2023, doi:10.3390/diagnostics13152601_

Round 1

Reviewer 1 Report

very interesting and well done research, it will be usefull for clinicians due to the clarity of the images

Author Response

Thank you so much for your review and comments. We highly appreciate your time and effort.

Reviewer 2 Report

My comments are:

  • The title is misleading. How is dysfunction of the infraspinatus fascia exclusively determined? More factors could intervene, among others, biomechanical variables.
  • I do not fully understand the clinical contribution of the paper. The diagnosis of the pathology is based on clinical manifestations such as snapping.
  • The authors provide many images to help understand the paper. But they should better describe the patient's baseline state and improvement, in relation to variables.
  • The authors show a video where the patient treats himself. I know the patient is a doctor but he should have been operated on by a colleague.
  • The authors show a video where the patient treats himself. I know the patient is a doctor but he should have been operated on by a colleague.
  • It is not clear to me, what is the work methodology? what infiltrates, how many sessions, additional treatment...

Author Response

    1. The title is misleading. How is dysfunction of the infraspinatus fascia exclusively determined? More factors could intervene, among others, biomechanical variables.

    Response: Thank you so much for your review and comments. We highly appreciate your time and effort. The title was chosen to reflect the final causes of this patient’s pain and its resolution. The patient is an experienced musculoskeletal physician who developed anterolateral shoulder pain and snapping. Multiple conservative treatments failed to stop the snapping and pain. A direct ultrasound-guided needle contact to the cortical break over the humeral head reproduced the exact pain, but did not stop the snapping or restore flexion power.  Investigation of an observed scapulothoracic dyskinesia revealed dysfunction of the infraspinatus fascia, evidenced by musculoskeletal examination, dynamic ultrasound-guided sonopalpation, and digital palpation. Subsequent ultrasound-guided injection of these suspected infraspinatus fascial defects with hypertonic dextrose immediately relieved pain and snapping and restored full functionality, confirming the diagnosis.

    1. I do not fully understand the clinical contribution of the paper. The diagnosis of the pathology is based on clinical manifestations such as snapping.

    Response: Thank you for your comment.  The snapping was one clinical manifestation, accompanied by concurrent clinical symptoms, but pathologic determination was based on a combination of dynamic ultrasound, ultrasound-guided sonopalpation with the transducer, ultrasound-guided digital palpation, and complete relief of the anterolateral pain and relief of the anterior shoulder snapping after ultrasound-guided injection of different suspected lesions of the observed infraspinatus fascial defects. This manuscript clearly showed how to perform clinical assessment and ultrasound diagnosis of the infraspinatus fascia and its related sonoanatomy. To our knowledge, this is the first literature to clearly delineate in detail the sonoanatomy of the infraspinatus fascia and its related muscles and bony landmarks. It is also important to point out that pain imitation by diagnostic injection does not confirm that the structure being injected is the source of pathology. In this case, we suspect but cannot confirm that the cortical break was secondary to the scapulothoracic dyskinesia. Ab-normal force transmission to the area due to altered mechanics of humeral head movement could be responsible for lesions development and sensitization of nerves in that area, such that injection would be painful and imitative. By relieving dyskinesia, we suggest that abnormal force transmission would have ceased. As clinicians, in addition to treating presenting symptoms and signs of the consequence (in this case the cortical breaks of the anterior humeral head), it is imperative to look for and treat the root causes of the consequence (IF lesions which caused scapulothoracic dyskinesia) to provide long-term relief and prevent recurrence.

    1. The authors provide many images to help understand the paper. But they should better describe the patient's baseline state and improvement, in relation to variables.

    Response: Thank you for your comment.    We added in lines 44-47 the following: “Substantial dysfunction and impairment were evidenced by 80% pain and 70% disability subscores on the Shoulder Pain and Disability Index (SPADI), and a total score of 73.8%, with a maximum severity of 100.  His pain and dysfunction were unresponsive to all manner of conservative musculoskeletal treatments, substantially affected his sleep, and vocation as a musculoskeletal physician.” Other detailed baseline conditions were further demonstrated by video 1, video 2, figure 1, and video 3. The initial treatment of the patient by ultrasound-guided intralesional injection of the anterior shoulder in video 4 did not release the snapping and did not improve flexion power of the shoulder with the palm facing up. Further musculoskeletal testing focused on scapulothoracic movement and the factors affecting it (Video 5).  Loss of tension or pre-stress of the IF of this patient was shown in video 7. Ultrasound-guided diagnostic and therapeutic injection using hypertonic glucose mixed with lidocaine to different targets within the IF defects were demonstrated in videos 9-12. After treating the IF, the immediate effect on anterior shoulder snapping and scapulothoracic movement were described in videos 13-14.  Video 15 demonstrated normalization of the power of resisted shoulder flexion with the elbow straight, shoulder flexed to 90°, and humerus externally rotated (palm up).  We have also added lines 344 – 350, “Pain was eliminated and function normalized immediately after the first ultrasound-guided injection of 15 % dextrose mixed with 0.1% lidocaine into the infraspinatus fascial defects. The total SPADI score improved from 73.8 to 40% at 6-week post-injection follow-up. A second treatment of the residual fascial defect sites with 15 % dextrose mixed with 0.1% lidocaine was administered, which again eliminated pain and resolved residual functional limitations. The patient received no other treatment, but he has been paying particular attention to any postures which would lead to damage in the IF, e.g., prolong protracting the shoulders during treating his patients manually. The patient was pain free without snapping symptoms or functional limitations at 3, 6, 12 and 48 months of follow-up.”

    1. The authors show a video where the patient treats himself. I know the patient is a doctor but he should have been operated on by a colleague.

    Response: Thank you for your comment. We have substituted the video of the ultrasound-guided injection of the painful anterior shoulder with a new video in which another doctor was treating this patient in the second session. 

    1. It is not clear to me, what is the work methodology? what infiltrates, how many sessions, additional treatment...

    Response: Thank you for your comment. We have added lines 344 – 350, “Pain was eliminated and function normalized immediately after the first ultrasound-guided injection of 15 % dextrose mixed with 0.1% lidocaine into the infraspinatus fascial defects. The total SPADI score improved from 73.8 to 40% at 6-week post-injection follow-up. A second treatment of the residual fascial defect sites with 15 % dextrose mixed with 0.1% lidocaine was administered, which again eliminated pain and resolved residual functional limitations. The patient received no other treatment, but he has been paying particular attention to any postures which would lead to damage in the IF, e.g., prolong protracting the shoulders during treating his patients manually. The patient was pain free without snapping symptoms or functional limitations at 3, 6, 12 and 48 months of follow-up.”

Reviewer 3 Report

Well presented 

Author Response

(The authors gave the same response as above.)

Round 2

Reviewer 2 Report

authors must establish an adequate structure of the paper. the paper must have a higher scientific level

Author Response

Response: Thank you for your comment. We have made significant changes to the structure of the paper to improve its flow and treatment strategy.

We add Lines 41 to 46, " This report aimed to present a case of painful anterior shoulder snapping, apparently due to compression of thickened fibrotic bursa between the subscapularis and the short head of biceps during external and internal rotation of the humerus. However, ultrasound-guided injection of the thickened and fibrotic bursa did not reproduce the pain or improve snapping. Ultrasound scanning identified a cortical break of the humeral head underneath the subscapularis as a potential mechanical effect of snapping. Nevertheless, ultrasound-guided injection of the cortical break reproduced and temporarily reduced the pain with snapping and residual anterior shoulder pain, but did not alter snapping or weakness in the right shoulder flexion with the humerus in external rotation.” And line 54 to 62, “Resisted shoulder flexion power tested by the treating physician with the elbow straight and the shoulder flexed at 90° with the internally rotated humerus (thumb down, similar to the empty can sign), both straight and abducted shoulder, had normal power and no pain. However, the power of resisted shoulder flexion with the elbow straight and the shoulder flexed to 90° with the humerus externally rotated (palm up) was diminished and associated with pain (Video 1).” 

We add line 70 to 72, “Dynamic ultrasound scanning of the painful anterior shoulder snapping during internal and external rotation was shown in Video 2. The cortical break underneath the subscapularis, suspected due the mechanical effects of the snapping and also causing the anterior shoulder pain was demonstrated in Figure 1. Ultrasound-guided digital palpation of the cortical break reproduced the patient’s usual pain in the anterolateral shoulder.”

Some of the content of the legends were moved to main body for better flow, lines 91 to 94 “Dynamic ultrasound scanning of the anterior shoulder with the humerus externally rotated (palm up), elbow straight and the shoulder actively flexed and abducted at about 90 to 100 degrees, the fibrotic and thickened bursa was noted to be snapping between the subscapularis tendon and the coracoacromial ligament as shown in Video 3. Patient noticed this snapping during many of his daily activities with his shoulder flexed just above 90 degrees, e.g. taking off clothes and combing hair.”

We have added lines 102 to 105, “Ultrasound-guided injection of the thickened and fibrotic bursa did not reproduce the usual pain nor reduce the pain and did not improve shoulder snapping and flexion power with the elbow straight, shoulder flexed to 90°, and humerus externally rotated (palm up). Ultra-sound-guided injection of the cortical break reproduced the concordant pain and temporarily and partially reduced the pain with snapping and residual anterior shoulder pain (Video 4), but did not change snapping or weakness in the right shoulder flexion with the humerus in external rotation.” Before video 4 to make it more clear.

We have added lines 118 to 125 to illustrate why need to do Video 5, “With the failure of direct anterior treatment to the suspected lesions, it was essential to further explore other causes for the anterior painful snap-ping. Further detailed musculoskeletal examination of the scapular movement showed that the right scapulothoracic movements were not smooth compared to the left counterpart (Video 5)[4, 5]. There might be some disruptions in the right infraspinatus fascia (IF) and its related muscles, including the lateral edge of the right inferior trapezius, rhomboid minor and major[6], teres major, latissimus dorsi [7, 8](LD), and posterior del-toid attachments to the IF [9, 10]. Clinically, prominent and active trigger points were observed in the right infraspinatus muscle. Holding the inferior angle of the scapula, with the examiner’s fingers, especially over the origin of teres major muscle over the scapula, the latissimus dorsi [11-13] and the scapular insertion of the rhomboid major muscle significantly improved the power of resisted shoulder flexion with the elbow straight and the shoulder flexed to 90° with the humerus externally rotated (palm up).”

We changed the legend of Video 5 slighlty, “Video 5. This video showed detailed musculoskeletal examination of the scapular movement with the right scapulothoracic movements com-pared not as smooth as the left counterpart[4, 5]. Still images of the suspected lesion sites were embedded in the video with labelling. Disruptions in the right infraspinatus fascia (IF) and related muscles were suspected, including the lateral edge of the right inferior trapezius, rhomboid minor and major[6], teres major, latissimus dorsi [7, 8](LD), and posterior deltoid attachments to the IF [9, 10].”

We added lines 141 to 144 to explain why we need to do so many videos demonstrations of scanning on normal individual about the infraspinatus fascia. And why need to have dissection pictures 6-8. “In order to illustrate the utilization of ultrasound-guided sonopalpation and ultrasound-guided digital palpation for detecting the suspected lesions of the IF and its related muscles, we presented the following videos (Videos 6 to 9) and still images (Figures 2-5) to demonstrate the essential techniques and crucial points of getting the normal sonoanatomy of IF and its related structures. The gross anatomy of IF and its related muscles were also shown in Figure 6-8.

We added lines 278-285 to illustrate the necessity of sonopalpation on diagnosis of IF defects: “Sonopalpation is to use the transducer to evaluate myofascial integrity. The aim of sonopalpation is to demonstrate any loss of tension or pre-stress of the fascia, IF in this patient (Video 10), usually seen as excessive displacement of the soft tissues when the transducer is compressed onto the soft tissue and the force is transmitted more easily to the bony attachments of IF. This may cause the scapular edge to excessively protrude toward the subcutaneous or even the dermal layer underneath the transducer. The decreased tension was interpreted as a disruption of fascia integrity, or compromised biotensegrity as described in the thoracolumbar fascia [14]. There is another alternative technique called ultrasound-guided digital palpation which serves to diagnose the loss of tension or excessive displacement of the fascia, which may signify fascial defects, as illustrated in Video 11 as an example. Both techniques have the advantage of real time comparison with the sound side as control.”

We added lines 341 to 347 to illustrate why we need to do the diagnostic and therapeutic injection of hypertonic glucose for the suspected IF defects. “In this patient, to prove the suspected lesions in IF found by ultrasound-guided sonopalpation, where it showed decreased tension and increased displacement of the soft tissue under sono-palpation compared to the sound side and partly shown on video 10, we performed ultrasound-guided injection of 15% hypertonic-glucose mixed with 0.1% lidocaine for the diagnostic and therapeutic in-jection. We injected the suspected lesions in the IF over its attachments with the teres major and LD on the inferior edge of the IF (video 12); in the lateral edge of the inferior trapezius attachments to medial side of the IF (Video 13 and Video 14); and in the superolateral attachment of the IF above the teres minor and major (Video 15). Hypertonic glucose is the primary injectate of these diagnostic and therapeutic injection, which postulates to induce the wound healing cascade and leads to tissue repair[15-17].

We have reorganized the videos 10 to 18.

We have redone the legends of Videos 12-15.

We added lines 374 to 378, “After the diagnostic and therapeutic hypertonic glucose injection, the volume of the injectate will temporarily increase the tension around the IF, which temporarily makes improvement of scapular dyskinesia and makes shoulder movement more efficient (video 16). In this patient, the shoulder snapping (Video 17) and his anterolateral shoulder pain were eliminated immediately after the injection of the above muscle attachments to the IF. Besides, the power of resisted shoulder flexion with the elbow straight and the shoulder flexed to 90° with the humerus externally rotated (palm up) was normalized (Video 18).” to point out the effects of our diagnostic and therapeutic injection and also show the importance of Videos 16 to 18 to show the efficacy of our treatment.

We redo the lines 486 to 490 before the conclusion to show the logic of failure of anterior shoulder injection to ease the anterior shoulder pain but the posterior injection of the IF could reduce the anterior shoulder pain and snapping by treating the dyskinesia.

Round 3

Reviewer 2 Report

Changes made by the authors have improved the paper.